# Bacterial Membranes Are More Perturbed by the Asymmetric Versus Symmetric Loading of Amphiphilic Molecules

**DOI:** 10.3390/membranes12040350

**Published:** 2022-03-22

**Authors:** W. F. Drew Bennett, Stephen J. Fox, Delin Sun, C. Mark Maupin

**Affiliations:** 1Physical and Life Sciences Directorate, Lawrence Livermore National Laboratory, Livermore, CA 94550, USA; sun25@llnl.gov; 2Procter and Gamble, Reading Tech Center, 452 Basingstoke Rd, Reading RG2 0RX, UK; fox.sf@pg.com; 3Procter and Gamble, Mason Business Center, 8700 Mason Montgomery Rd, Mason, OH 45040, USA; maupin.cm@pg.com

**Keywords:** bacteria inner membrane, membrane asymmetry, molecular dynamics simulations, lateral pressure profiles

## Abstract

Characterizing the biophysical properties of bacterial membranes is critical for understanding the protective nature of the microbial envelope, interaction of biological membranes with exogenous materials, and designing new antibacterial agents. Presented here are molecular dynamics simulations for two cationic quaternary ammonium compounds, and the anionic and nonionic form of a fatty acid molecule interacting with a *Staphylococcus aureus* bacterial inner membrane. The effect of the tested materials on the properties of the model membranes are evaluated with respect to various structural properties such as the lateral pressure profile, lipid tail order parameter, and the bilayer’s electrostatic potential. Conducting asymmetric loading of molecules in only one leaflet, it was observed that anionic and cationic amphiphiles have a large impact on the *Staphylococcus aureus* membrane’s electrostatic potential and lateral pressure profile as compared to a symmetric distribution. Nonintuitively, we find that the cationic and anionic molecules induce a similar change in the electrostatic potential, which points to the complexity of membrane interfaces, and how asymmetry can induce biophysical consequences. Finally, we link changes in membrane structure to the rate of electroporation for the membranes, and again find a crucial impact of introducing asymmetry to the system. Understanding these physical mechanisms provides critical insights and viable pathways for the rational design of membrane-active molecules, where controlling the localization is key.

## 1. Introduction

The structure and stability of lipid membranes and the effect of adding exogenous molecules to the membrane is important for many biological processes, and biotechnology. Lipid membranes surround every cell, most cellular organelles, and even many viruses, in addition to forming skin layers and the blood–brain barrier. The complexity of membrane systems cannot be overstated, with interconnected processes spanning time scales from picoseconds to hours, and size scales from single molecules/atoms to macroscopic aggregates. Surrounding a cell with a nanometer-thin, low-dielectric slab allows for energy production/conversion, nerve propagation, and electrochemical-based molecular transport, in addition to many other crucial life processes. To support these numerous functions, cells tightly control the composition of each membrane, with tens of thousands of unique lipid species [1]. Decades of biological membrane research has shown that many cell membranes also display asymmetric lipid distributions and lateral membrane domains, both of which are key to many biophysical mechanisms, such as virus entry/exit, bacterial infection, cancer, inflammation, and cell signaling [1,2].

Bacterial membranes are compositionally distinct from mammalian membranes, for example, bacteria contain more negatively charged phospholipids and no cholesterol, compared to mammalian cells [1]. This lipid composition difference has important consequences for the biological properties of cells, for example, dictating what can passively diffuse through the membrane. Additionally, the membrane composition affects energy production in bacteria. Characterizing how membranes respond to external stimuli, such as the addition of molecules, is crucial for many biotechnological applications. Many surfactants and antimicrobial peptides kill bacteria through a membrane-mediated mechanism [3]. Strong detergents can completely lyse the bacterial membrane, while mild surfactants can have less drastic effects, but both are concentration-dependent. Similarly, antimicrobial peptides also have a range of function, from forming stable membrane pores to more subtle effects, such as disrupting the membrane’s electrostatic potential [4]. Of note, for this work, fatty acids [5] and alkyl-quaternary amine amphiphiles [6] have been shown to have antibacterial activity. Understanding how surfactants affect bacterial membranes is critical to formulating mixtures that are inherently hostile to microbial contamination and for designing new antibacterial molecules.

There have been many previous studies using molecular dynamics (MD) computer simulations to characterize bacterial model membranes [7]. Recent advances in simulation size and force field development have led to complex membrane simulations of bacterial membranes. For example, simulations have recently been used to study bacterial cytosolic membranes, outer membranes, peptidoglycan layers, and very large simulations of the outer membrane, periplasmic space, and inner membrane [7,8,9,10]. Beyond basic membrane biophysics, there are many simulation studies on bacterial membrane proteins, which require an accurate model for the lipid membrane [11,12]. There has also been a very large number of papers simulating antimicrobial peptides [13]. Recent advances include using free energy calculations for antimicrobial peptides binding and penetrating bacterial model membranes.

There have also been many studies for small molecules moving through membranes, but these mostly focus on single molecules, not the effect on the membrane properties. Exceptions include cholesterol, and other sterols, which have been studied and reviewed extensively [14], and fatty acids, mostly oleic acid [15], which have relevance to this work, and have informed our present study. While adding amphiphilic small molecules can affect the structural properties of membranes, linking membrane structural changes to biological outcomes is challenging. Lateral pressure profiles measure the tension/stress at slices through the membrane and can be used to calculate membrane elastic properties and estimate transmembrane protein changes in conformational free energy [16,17]. Electrostatic potential changes calculated in simulations, from the average charge distributions, can be compared to experimental estimates. Adding a large potential difference across a membrane causes a hydrophilic pore to form across the membrane, which has been studied extensively with atomistic simulations [18,19,20,21].

We report results from detailed atomistic simulations of neutral fatty acid, negatively charged fatty acid, and two positively charged quaternary ammonium molecules interacting with a model *Staphylococcus aureus* (*S. Aureus)* inner membrane. Using atomistic MD simulations with extensive structural characterization, we explore the effect of small-molecule insertion, and the difference between inserting in both leaflets and asymmetrically in only one leaflet. Collective structural changes are characterized with lateral pressure profiles, partial density profiles, and electrostatic potentials. These changes are linked to individual lipid conformation changes upon insertion of different molecules. Finally, electric fields are applied to the systems to investigate how structural changes affect hydrophilic pore formation. We observe large structural changes, particularly when the charged molecules are in only one leaflet. These results have implications on future antimicrobial molecular design, where molecules that do not flip to the inner cytoplasmic leaflet would be more effective.

## 2. Methods

### 2.1. Simulation Protocol

CHARMM36 lipids [22] with CHARMM-specific [23,24] TIP3P water [25] were simulated with GROMACS v2018.3 [26,27]. For the *S. aureus* inner membrane, we used a 54 mol% di-anteiso-myristoyl-phosphatidylglycerol (AMPG), 36 mol% di-anteiso-myristoyl-lysyl-phosphatidylglycerol (ALPG), 10 mol% di-phosphatidylglycerol (or cardiolipin) (ADPG) (-ve 2 protonation state and four anteiso-myristoyl chains) lipid mixture that was built based on previous literature [28]. The anteiso-myristoyl chains are a 14-carbon saturated chain with a methyl group on the second to terminal carbon. Each bilayer leaflet contained 100 lipids. Starting with the pure *S. aureus* membrane, we inserted amphipathic molecules to assess how the bacterial membrane properties are affected by different molecules. The molecules studied were an anionic deprotonated lauric acid (LAU) (29 mol% or 40 molecules in each leaflet), a cationic C16-diethyl ester dimethyl ammonium (BQT) (17 mol% or 20 molecules in each leaflet), a cationic C12-tri-methyl ammonium (LQT) 29 mol%), and a neutral, protonated lauric acid (LAUP) (29 mol%). These molecules were parameterized using the CGENFF [29], and the chemical structures are shown in Figure 1. We chose roughly twice as many fatty acids and LQT compared to the BQT because they have only one acyl chain, compared to two for BQT. We also tested the effect of adding molecules to only one monolayer of the *S. aureus* membrane. Given that charged molecules are likely to undergo flip-flop on a slow timescale, the asymmetric addition of the charged molecules is a likely scenario for exogenous materials interacting with bacteria membranes. In this case, BQT, LQT, and LAU molecules were only inserted in the lower leaflet of the *S. aureus* membrane (i.e., the leaflet on the negative z-axis compared to the center of the membrane). For the asymmetric bilayers, we also simulated double bilayer systems, with Figure 2 showing an example of the bilayer setup and the molecules added to the ‘outer’ and/or ‘lower’ leaflet.

Simulations were run using a 2 fs time step, with 4 replicates of 200 ns, with the final 150 ns from each used for analysis (600 ns per system in total). Lennard–Jones interactions were cut-off after 1.2 nm, with a force-switch-function from 1.0 to 1.2 nm. Long-range electrostatic interactions were calculated with the particle mesh Ewald method [30,31]. The temperature was maintained at 313 K using the Nose–Hoover thermostat [32] with a time constant of 1 ps. Semi-isotropic pressure coupling was used to maintain a constant pressure of 1 bar lateral and normal to the plane of the bilayer. The Parrinello–Rahman barostat [33] was used with a compressibility of 4.5 × 10^−5^ and a coupling constant of 5 ps. Water was constrained with the SETTLE method [34], and hydrogens on lipids with P-LINCS [35,36].

For analysis, we used MDAnalysis [37] for in-house-developed density and electrostatic potential calculations. Area per lipids was calculated using FatSlim [38], which uses Voronoi tessellation. The atomistic lipid tail order parameter was calculated using:$$S_{CH}=<3\;cos^{2}\;\theta -1>/2$$
where *θ* is the angle between the C-H bond and the normal to the bilayer. This was calculated using an analysis script in MDAnalysis [37,39,40]. The bilayer compressibility modulus was calculated using:$$K_{A}=kT<A>/<N{\left(A-A_{0}\right)}^{2}>$$
where *A* is the instantaneous bilayer area, *A*_0_ is the average bilayer area, and *N* is the number of lipids in each bilayer leaflet. Due to the different number of lipids in each leaflet for the asymmetric systems, we did not calculate compressibilityfor these systems.

The lateral pressure profile was calculated using a modified version of GROMACS [41,42]. The local stress was calculated by post-processing the trajectory file, which returns a local stress file for the simulation averaged over the user-selected time range. We used the Goetz–Loetsky force decomposition and a grid spacing of 0.1 nm for atomistic. The local stress tensor (σ) was extracted from the binary file using the program tensortools [41,42].

### 2.2. Voltage-Induced Pore Formation

Using the same run parameters but with a constant applied field, we ran simulations for the *S. aureus*, the symmetric BQT, LQT, and the asymmetric BQT and LQT membrane systems. For each membrane, we ran 24 replicates with a field strength of 0.25 V/nm and 24 with −0.25 V/nm. Simulations were monitored for pore formation, judged by the spike in area per lipid immediately after pore formation. The simulations were run up to 60 ns. The field strength was chosen based on literature values and a short trial and error test to observe pores on a reasonable simulation time scale. The probability for pore formation as a function of time was then calculated from the cumulative histogram of the pore formation times. The mean pore time and standard error were calculated, and the pore formation probability was fit with a stretched exponential curve [43,44].

## 3. Results

Figure 3 shows bilayer partial density profiles for each system, as a representative of total bilayer structural changes. The total lipid densities (including the added BQT, LQT, LAU, or LAUP) are shown in Figure 3A. As expected, the maximum density is at the head group region, with a trough near the bilayer center. Overall, the lipid density is very similar for the pure *S. aureus,* and when BQT or LQT is added. For LAUP compared to pure *S. aureus*, the density profile shows that the bilayer thickness is the same, as well as the trough at the bilayer center, but the head group density is reduced. The negatively charged LAU causes the most significant changes to the density profile, causing considerable thinning of the bilayer, a reduction in density at the head group region, and a slight increase in density at the bilayer center.

Figure 3B shows the bilayer density for the asymmetric membranes, with the molecules added to only the lower leaflet, as shown in Figure 2A. These profiles show how when molecules are added to only one leaflet, the entire bilayer density is changed, with the overpopulation in the lower leaflet causing underpopulation and a lower density in the opposite leaflet. The trough at the bilayer center is only very slightly shifted compared to the symmetric *S. aureus* membrane. These shifts in density match trends in the change in the area per lipid data in Table 1, where each molecule causes a decrease in each type of lipid in the *S. aureus* membrane. The largest decrease is for the LAU system where the cardiolipin ADPG’s area per lipid decreases from 0.83 to 0.64 nm^2^. This large decrease is likely due to electrostatic repulsion between LAU and the di-anionic ADPG. The change in areas per lipid for the asymmetric lipid system also matches the shifts in partial density observed in Figure 3B, with the lower leaflet decreasing and the upper leaflet increasing to compensate for the lower number of molecules.

The elastic properties of lipid membranes have important structural and functional consequences for cells. Table 1 shows the effect of a cationic, anionic, and neutral amphiphile on the compressibility modulus for the *S. Aureus* lipid bilayer. The cationic LQT and BQT molecules and the neutral LAUP molecule cause a small increase in the compressibility (Table 1), while the negative LAU molecule reduces the compressibility of the membrane significantly.

To assess the molecular properties that are giving rise to the changes in membrane structural and elastic properties, we calculated the order parameter for the lipid acyl chains. The order parameter (S_C–H_) measures the angle between the vector connecting a carbon and hydrogen on the lipid chain and the *z*-axis (normal to the plane of the membrane). This measures how aligned the lipid chains are, with a larger order parameter indicating that the chains are more aligned with the bilayer normal. Figure 4A shows the order parameter for each carbon in the *sn*-2 tail of AMPG lipids in each system. BQT, LQT, and LAUP cause an increase in order parameter for the AMPG tails, while LAU causes a slight decrease in the order (Figure 4A). The shifted order parameters show how lipids adapt their conformation to different environments. For example, with BQT or LQT, the overall bilayer density does not change, but each lipid molecule is affected, and their molecular properties are changed. The increase in lipid tail order can also be linked to the increased compressibility for the BQT and LAUP bilayer. For the asymmetric membranes, we plot the lower and upper leaflets separately (Figure 3B), showing how the lower leaflet becomes more ordered, while the upper leaflet is less ordered.

The lateral pressure profile is an important measure for the tension/strain of the membrane. For example, pressure profiles are linked to the functioning of transmembrane proteins, as the membrane stress directly influences the free energy for protein conformational changes [16]. Figure 5A shows the pressure profiles across the *S. aureus* membranes with and without the tested materials. Given the large error in these calculations, the LAUP, BQT, and LQT have no significant effect on the membrane’s pressure profile. LAU causes substantial changes in the pressure profile, with a reduced depth of the pressure trough at the lipid/water interface, a slight reduction in the peak at ~+/−1 nm, and a drastic reduction at the bilayer center. These significant changes in the pressure profile could have important consequences on the bacterial membranes biological processes. These changes are also likely linked to the large reduction in compressibility for LAU.

Figure 5B shows the lateral pressure profiles for the *S. aureus* membrane with BQT, LQT, and LAU in only one leaflet. As noted above, the symmetric addition of BQT or LQT causes little change in the pressure profile compared to *S. aureus.* The asymmetric BQT or LQT has a significant effect, with large changes occurring around +/−1 nm, which is the tail region of the membrane, although the other features are not significantly altered. Despite the vast difference in chemical structure, BQT and LQT cause very similar changes to the pressure profile when added asymmetrically. For LAU, an asymmetric distribution also has significant changes, but to all regions of the pressure profile, characterized by mostly a reduction in the magnitude of the pressures.

Figure 3C shows the electrostatic potential across each lipid membrane. As there is no charge imbalance across these membranes and the systems are charge-neutral, the electrostatic potential changes across the membrane are due to dipoles aligning at the membrane interface. There is a positive potential at the center of the lipid membranes, as expected from previous simulations and experiments. Adding LAUP slightly reduces the potential at the bilayer center compared to pure *S. aureus*. LAU causes the membrane to have a significantly lower electrostatic potential compared to pure *S. aureus*. For BQT, the membrane potential is the same as *S. aureus,* despite the net positive charge on BQT. The electrostatic potential profiles across the membrane with BQT, LQT, and LAU in only one leaflet are shown in Figure 6. For this calculation, we used a double bilayer setup so that we can have two unique aqueous compartments, one adjacent to the added molecules (outer) and the other opposite (inner). We should note that counterions were added to ensure there is no charge imbalance across the membranes. The asymmetric addition of either BQT, LQT, or LAU causes a drop in the electrostatic potential in the inner water compared to the outer water of ~120 mV.

In atomistic simulations, the electrostatic potential is derived from the average charge distribution. To gain insight into the source of the potential differences, we also calculated charge distributions for individual molecule groups, which are shown in Figure 6C–E. We note that we are showing distributions for only one of the bilayers for clarity, but the other bilayers distribution was the same. The magnitude of the total charge distribution is relatively small (<+/−0.2 e) compared to the individual components (Figure 6C). Water’s charge distribution is zero in water due to the homogeneity of bulk water and isotropic dipole orientation and has a negative trough and positive peak at the membrane interface. This distribution is caused by water molecules aligning with their oxygen atoms toward water and the hydrogens toward the bilayer center. Figure 6D shows the lipid charge distribution including either BQT or LAU, in only one leaflet. The membrane lipids have a large negative charge distribution at the head group region, due to the net negative charge on the ADPG and AMPG lipids. BQT has a large positive peak, and LAU a negative peak at the interface, as expected. The lipids in the LAU system shift substantially, while for BQT, the lipids display a modest change in charge distribution. In Figure 6E, the mobile ion charge distributions are shown, with Na^+^ concentrating on the membrane interface. There are major changes to both Na^+^ and Cl^−^ distributions when BQT or LAU is added to one leaflet. Finally, we note that the charge distributions change substantially in the leaflet on the opposite side of the added BQT or LAU, illustrating how the membrane adapts to asymmetric structural changes.

As a test for how our observed structural changes can affect crucial membrane integrity, we simulated membranes with an applied electric field in the direction normal to the plane of the membrane. Electroporation has been used extensively to understand membrane pores [18,19,20,21,45], which we used to select ~0.25 V/nm as a reasonable field. We tested a few different fields (data not shown), settling on 0.25 V/nm as a balance for the different membranes observing poration on a time scale we can simulate. Higher voltages result in nearly instantaneous pore formation. Figure 7A,B show snapshots of the asymmetric BQT membrane before and after pore formation, matching previous simulations. By running 24 replicates and measuring the time for a pore to open, we can plot the probability for a pore to form at a given time (Figure 7C). Similarly, we list the mean time to form a pore from the 24 replicates in Table 2.

From the noise in raw data in Figure 7D and the difference in mean pore time for symmetric bilayers, 24 replicates are inadequate to obtain reliable statistics. The broader trends that we observe are very clear. Inserting BQT or LQT molecules in both membrane leaflets reduces the chances that a pore will form. While noisy, this matches the increased compressibility moduli in Table 1, i.e., stiffer membranes are less likely to form pores. The asymmetric addition of BQT results in interesting behavior with respect to applying a positive versus negative potential. The positive field results in significantly more pores, while the negative field has less pores than the pure *S. aureus* membrane (which has the same lipid composition). Given that BQT causes an electrostatic potential difference (Figure 6B), this could explain how the applied field can be enhanced or compensated by this intrinsic potential difference. We note that the applied field results in a potential of 2.4 V across the box, which is much larger than the ~0.1 V intrinsic dipole potential in the asymmetric systems. While the asymmetric addition of the molecules weakens the mechanical strength (Table 1), with the −0.25 V/nm field, pore formation is also reduced, but to a lesser extent. This suggests that compressibility alone cannot explain the effect of pore formation, as the direction of the field causes opposite behavior. For LQT, we observe similar pore formation rates with the −0.25 V/nm and +0.25 V/nm (Table 2 and Figure 7C). Sorting out these differences will require more replicates and in-depth analysis.

## 4. Discussion

Our results show several interesting and nonintuitive outcomes of adding small amphiphilic molecules to a bacterial membrane. Adding molecules to one side or both can have vastly different and even opposite effects on membrane properties. For BQT, the membrane becomes stiffer when added to both leaflets, likely due to strong interactions between the cationic BQT and the anionic AMPG and di-anionic ADPG molecules. When added to only one leaflet, the membrane becomes much less stiff, even compared to pure *S. aureus*. Adding molecules to only one leaflet causes the system to become asymmetric, but overall, there is still a tendency for the system to accommodate this change. This behavior is exemplified by the changes in properties we observe in the leaflet opposite to the added BQT molecules. Due to the expanded volume of BQT molecules on one leaflet, lipids on the opposite leaflet are forced to occupy a larger area per lipid (Table 1). This stress shows up in the shift in the lateral pressure profiles for the asymmetric systems (Figure 5B). The cooperativity in real systems goes beyond our calculated properties, where every membrane protein’s structure and activity would be modified by these bulk membrane changes.

A key function of biological membranes is to maintain specific electrochemical gradients. The text-book example is nerve cells firing, where shifting ion concentrations flip the potential gradient, resulting in an action potential. Many bacteria use this gradient for energy production, molecule transport, and sensing the environment. Therefore, disrupting this gradient can cause bacterial cell death. Adding cationic or anionic molecules to only one side of the membrane causes a dipole potential difference between the two adjacent water layers. Of note, our results suggest that both cationic (BQT and LQT) and anionic (LAU) molecules shift the electrostatic potential in the same direction and by a similar magnitude (~120 mV), despite having opposite charges and twice as many LAU molecules than BQTs. It is of particular interest that both the positive and negative molecules on the outer leaflets cause an increased potential on the inner leaflet. The difference of ~120 mV is substantial given that the resting potential for bacterial membranes is estimated to be in the 100s-of-mV range (with the inside of the cell negative) [46]. Previous simulation studies have shown that asymmetric distributions of neutral lipids can also alter the electrostatic potential across the membrane [47]. Digging into the detailed charge distributions, we show that this behavior is due to vastly different mechanisms, and the similar magnitude is likely a coincidence. We stress the complex interplay and collective behavior of all the molecules/ions in the system. Breaking apart the charge distributions, we show the complexity of the underlying mechanism of this bulk behavior. Water, lipids, and mobile ions can all shift their location, orientation, and/or conformation when other charged species are inserted in one membrane leaflet. Teasing out the effect of the charged head group compared to the increased number of lipids on one side will require more research. We note that this is a nontrivial task, and especially if one is to include the effect of membrane bending. Our results, showing the asymmetric addition of the cationic molecules, cause changes to the electrostatic potential and do not increase pore formation propensity, matching results from a previous experimental study on cationic ceragenin antimicrobial molecules [48]. Given the large structural changes that we observe with LAU inserted both symmetrically and asymmetrically, it is likely that negative antibiotic molecules can have different mechanisms of action, compared to cationic molecules.

In addition to preventing bending, our simulations also preclude lipid flip-flop, which is another key mechanism that membranes use to alleviate membrane asymmetry. Simulations [49,50,51] and experiments [52,53] have shown that the rate for lipid flip-flop varies considerably depending greatly on head group chemical structure and charge state, and particular membrane environment. Even fast-flipping lipids are expected to flip on the millisecond timescale, well beyond our current simulation length. Longer flip-flop rates of hours are expected for amphiphilic molecules such as BQT, suggesting that a fully asymmetric distribution (on the outer leaflet) is a relevant state after mixing BQT with a *S. aureus* membrane. For the LAU/LAUP systems, we also neglect (de)protonation, whereby the fraction of each is linked to the solution pH outside and inside the cell. We also did not simulate the neutral LAUP molecule asymmetrically inserted in the membrane, because it can likely flip so fast that it would quickly equilibrate between the membrane leaflets. There have been several constant-pH studies of lipids and fatty acids [15,54], but they are still expensive computationally and difficult to set-up. Linking flip-flop, ion imbalances, and pH imbalances in simulations of large, multicomponent membrane systems will provide novel biophysical insights that have broad interest and important applications. We have also shown that the asymmetric distribution of amphiphilic molecules considerably alters the membrane lateral pressure profiles. Lateral pressure profiles quantify the compression/tension through the membrane, which is crucial for membrane properties, and the effect on transmembrane proteins. The free energy for a transmembrane protein to shift conformations, such as a mechanosensitive channel opening/closing, can be calculated from the lateral pressure profile and the change in membrane-exposed surface area of the channel. Our pressure profiles show how the asymmetric distribution of charged surfactants alters bacterial membrane properties, which would lead to adverse stress on the microbe and possible cell death. For the symmetric systems, only the negatively charged LAU molecules have a significant effect on the pressure profiles. The BQT and LQT molecules in one leaflet cause substantial changes to the tail region of the membrane’s lateral pressure profile (Figure 5). Despite 600 ns of sampling, these curves are still noisy, making a real quantitative assessment not possible.

We anticipate that future work correlating extensive computational predictions with experiments will allow quantitative structure–activity relationships but will require an extremely large computational effort. There has been a recent surge in experimental advancements for creating and studying asymmetric model systems, such as large unilamellar vesicles [55], in addition to novel lipidomics of individual membrane leaflets [56]. Future work will also include expanding the number and structure of molecules studied, in addition to testing if this behavior is similar in other model membranes, such as a model human plasma membrane. Additionally, exploring the effect of the force field and models that include polarizability will be of interest, given that the CHARMM36 force field we have used is a fixed-charge model [57]. Given the importance of membranes in biology, we expect many future studies in this area, with more realistic membranes, and aligned experiments for validation. To aid future work, data and structures will be provided upon request.

## 5. Conclusions

The interactions of molecules at an interface are complex, with strong driving forces, often in opposing directions. Adding charged molecules to a complex model of the inner membrane of *S. aureus* bacteria shows that the overall membrane properties can change substantially, as well as individual lipid molecular properties. The asymmetric addition of charged molecules has a large effect on the bilayer’s elastic and structural properties and can shift the potential across the membrane. Changing the membranes structural properties leads to changes in cooperative processes, such as pore formation after applying an electric field. Pores are less likely to form in membranes with amphiphiles in both leaflets and are equally probable without and with an asymmetric addition. These results suggest that some cationic amphiphiles kill bacteria by perturbing the membrane structure through their asymmetric distribution, and not by directly forming pores. Designing novel antibacterial molecules should consider the flip-flop rate and distribution across the bacterial membrane.

## Figures and Tables

**Figure 1 membranes-12-00350-f001:**
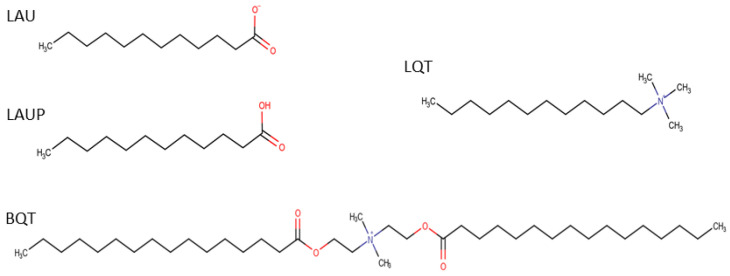
Chemical structures for the 4 molecules studied: laurate (LAU), lauric acid (LAUP), C16-diethyl ester dimethyl ammonium (BQT), and C12-tri-methyl ammonium (LQT).

**Figure 2 membranes-12-00350-f002:**
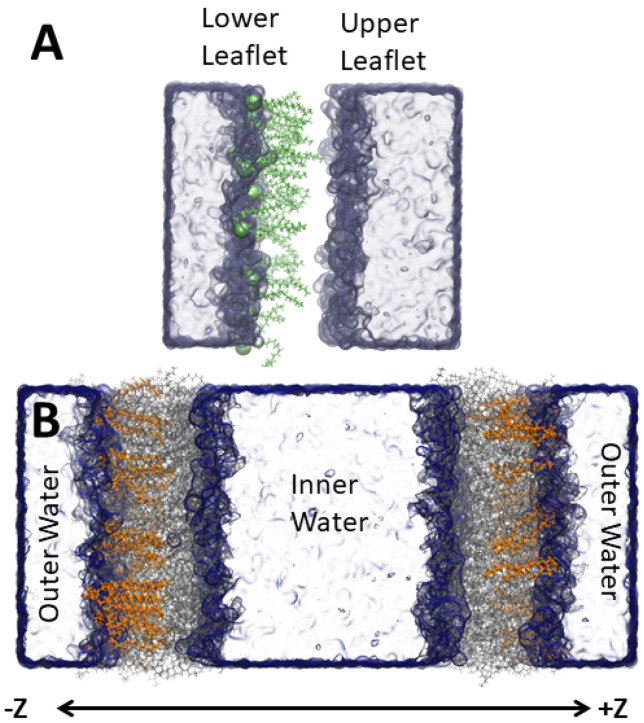
System setup illustrating the single (**A**) and double bilayers (**B**) with the amphiphilic molecules added to the ‘lower’ leaflet in the single bilayer and the ‘outer’ leaflet in the double bilayer. The added molecules are highlighted as green (**A**) and orange lines (**B**). The other lipids are grey lines, and water a blue surface.

**Figure 3 membranes-12-00350-f003:**
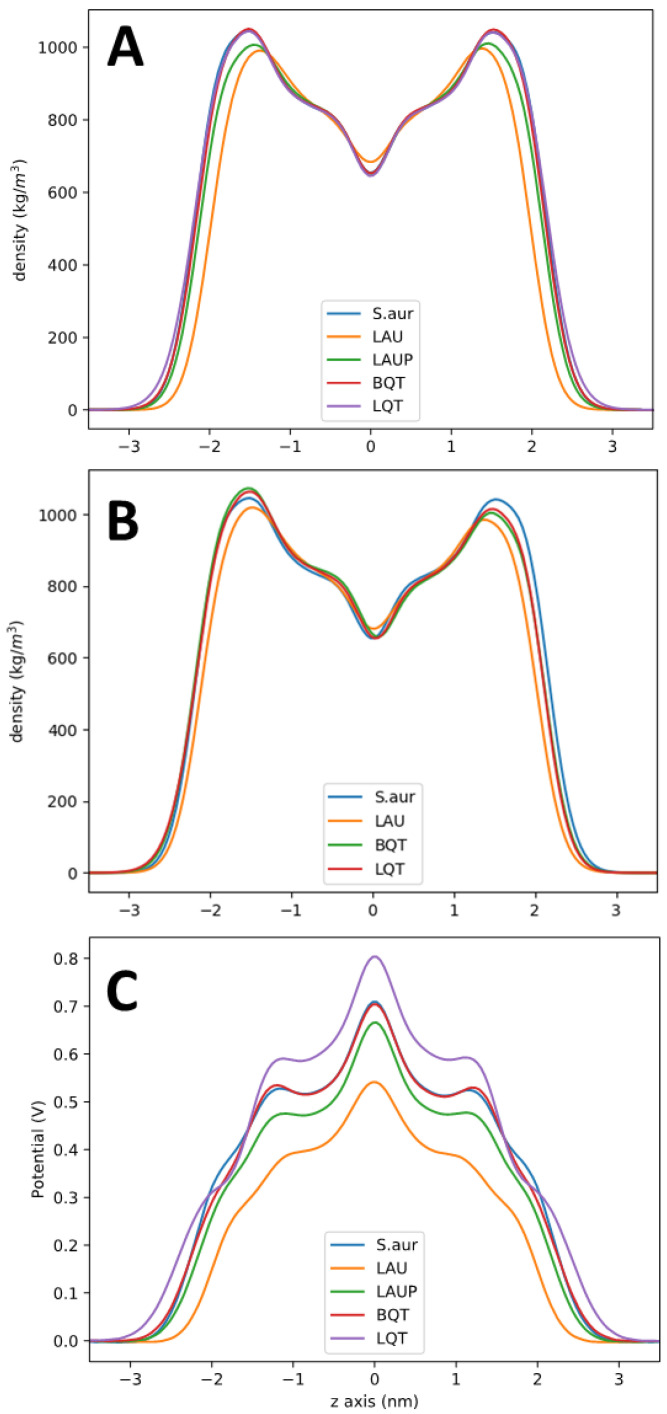
(**A**)**.** Partial density profile for pure *S. aureus*, and with LAU, LAUP, LQT, and BQT. The density is for all the lipids and small molecules in the system. (**B**)**.** Partial density profiles for the asymmetric membrane systems. (**C**)**.** Electrostatic potential across each membrane.

**Figure 4 membranes-12-00350-f004:**
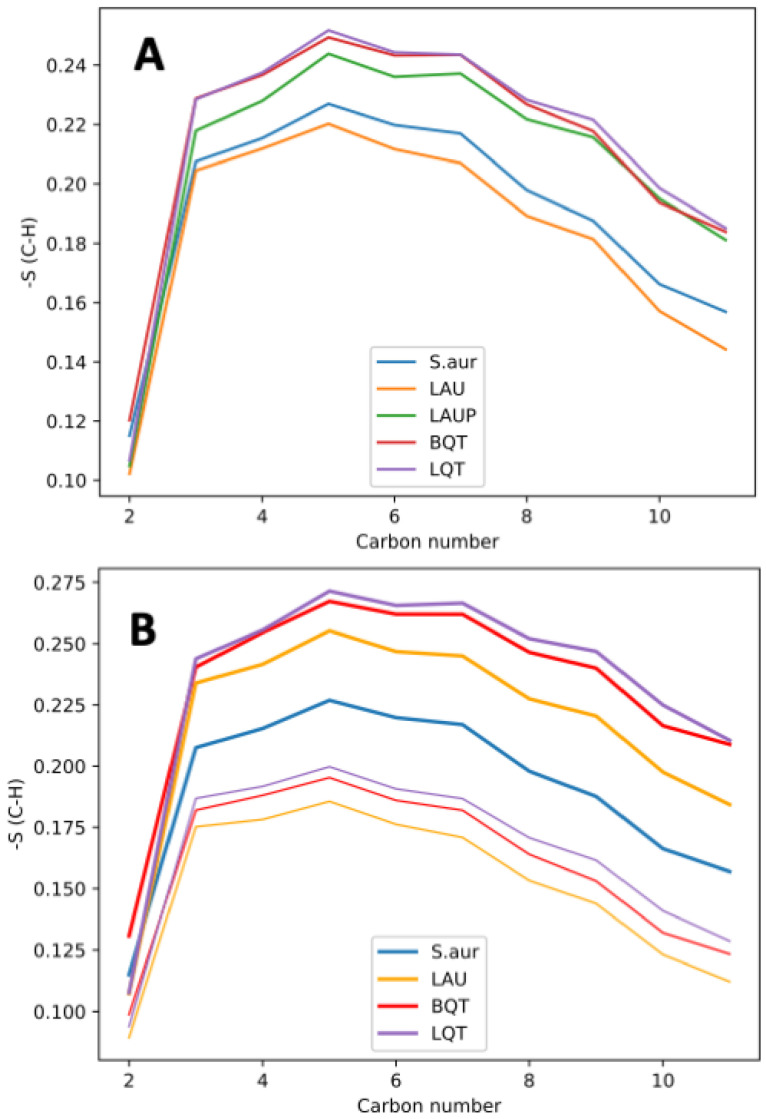
Order parameter for each carbon in the *sn*-2 tail of AMPG lipids in each membrane. (**A**) Symmetric membranes. (**B**) Asymmetric membranes with the enriched lower leaflet in thin lines and the empty leaflet in thick lines.

**Figure 5 membranes-12-00350-f005:**
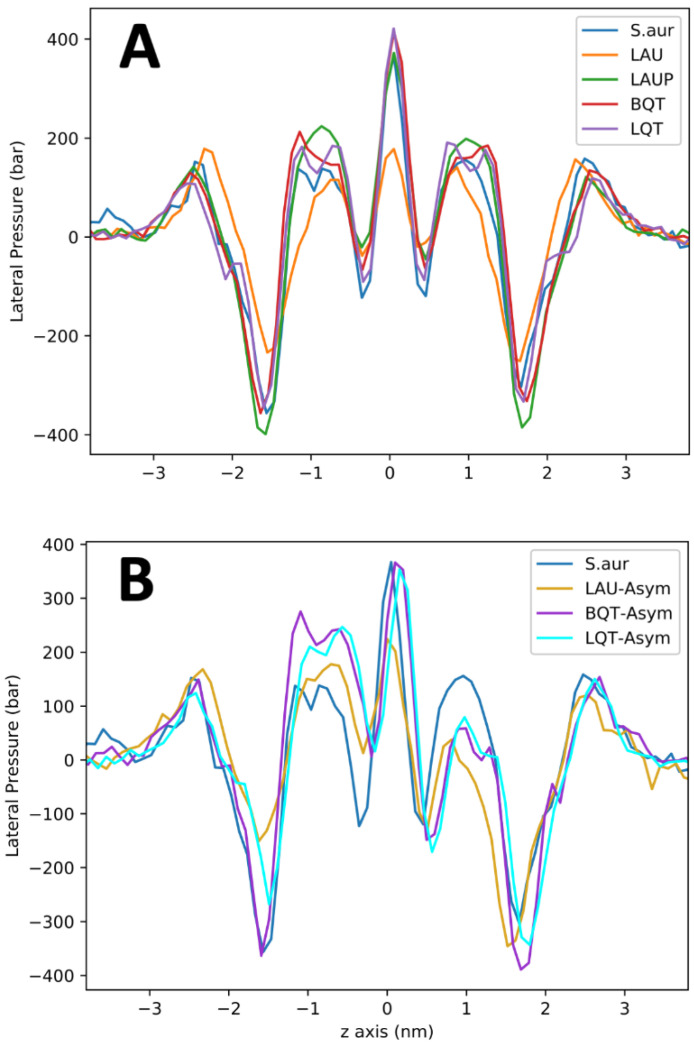
Lateral pressure profiles for symmetric (**A**) and asymmetric (**B**) distributions of molecules in the *S. Aureus* membrane. The asymmetric membranes contain the surfactants in the lower monolayer (i.e., -ve z).

**Figure 6 membranes-12-00350-f006:**
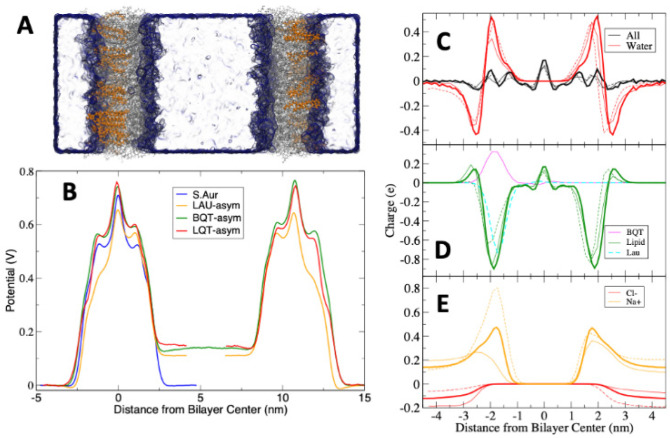
Electrostatic potentials across the *S. aureus* membranes with BQT and LAU in either one or both. For the asymmetric membrane, we used a double bilayer (see methods), ensuring the potential is equal at the box edge. (**A**) Snapshot illustrating the double bilayer, with the BQT surfactant highlighted as orange lines. The other lipids are grey lines, and water a blue surface. (**B**) Electrostatic potential profiles across the asymmetric double bilayers. For LQT and LAU, we cut the curve and shifted the right side to match the BQT bilayer for easier comparison. (**C**–**E**) Charge distributions for each system, *S. aureus* (thick lines), BQT (thin lines), and LAU (dashed lines).

**Figure 7 membranes-12-00350-f007:**
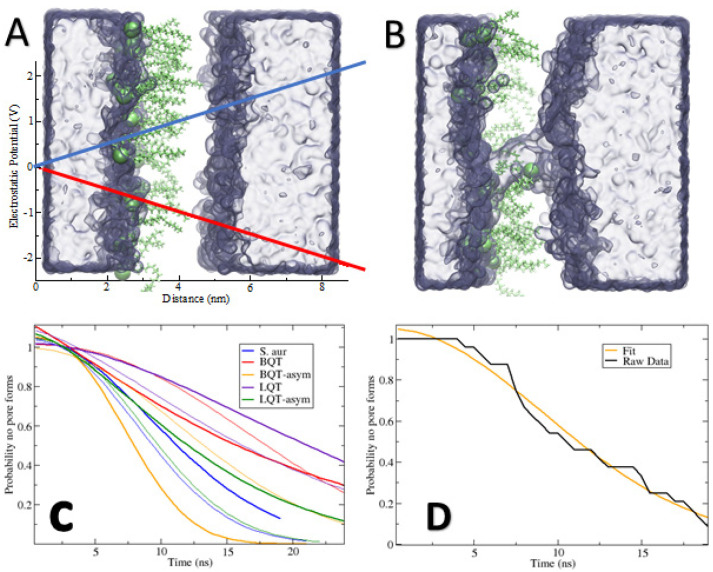
(**A**,**B**) Snapshots of an asymmetric BQT membrane system before and after pore formation due to the applied electric field. Water is a blue surface, and BQT is lime green, while the *S. aureus* lipids are not shown. The blue and red lines in A show the electric field strength across the membrane. (**C**) Probability that a pore will not form at a given time. The thin lines refer to systems with a negative applied field. Probabilities were from 24 replicates for each system and observing the time it takes for a pore to form, after fitting to a stretched exponential function. (**D**) Example showing the raw data and the stretched exponential fit for the *S. aureus* system.

**Table 1 membranes-12-00350-t001:** Areas per lipid and compressibility modulus for the membranes. For the asymmetric membranes, we report the areas for the upper/lower leaflets.

Membrane	K_A (mN/m) (std. Error)	Area Per AMPG (nm^2^)	Area Per ADPG (nm^2^)	Area Per ALPG (nm^2^)	Area of Molecule (nm^2^)
S. Aur	311 (32)	0.680	0.834	0.667	
LAUP	324 (23)	0.595	0.670	0.581	0.423
LAU	239 (23)	0.624	0.733	0.585	0.489
BQT	331 (17)	0.652	0.732	0.666	0.562
LQT	337 (22)	0.527	0.589	0.528	0.418
LAU asym		0.737/0.571	0.782/0.643	0.769/0.550	0/0.465
BQT asym		0.728/0.623	0.861/0.716	0.728/0.626	0/0.540
LQT asym		0.723/0.504	0.800/0.587	0.697/0.486	0/0.399

**Table 2 membranes-12-00350-t002:** Mean time (ns) for a pore to form in the different bilayers at the two opposite applied electric fields.

Bilayer	0.25 V/nm	−0.25 V/nm
S. Aur	11.8 (1.0)	10.6 (1.3)
BQT	19.3 (2.7)	21.0 (2.6)
BQT_asym	8.9 (0.9)	14.2 (1.6)
LQT	21.8 (2.1)	19.2 (2.5)
LQT_asym	13.9 (1.7)	12.3 (1.1)

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
