# Peer review of "Bacterial Membranes Are More Perturbed by the Asymmetric Versus Symmetric Loading of Amphiphilic Molecules"

_membranes, 2022, doi:10.3390/membranes12040350_

Round 1

Reviewer 1 Report

Comments on” Bacterial membranes are more perturbed by the asymmetric 2 versus symmetric loading of amphiphilic molecules”

This manuscript describes MD simulations of a model bacterial membrane perturbed by small amphiphilic molecules. The authors studied both anionic and cationic surfactants. The work offers new insight into this problem and is of a good technical level. I will recommend its acceptance into Membranes after the following issues have been resolved:

  • The text should be carefully proofread. For example, the Introduction starts from “T The structure and…” which gives a bad impression for further reading.
  • The introduction, there are many statements that are not supported by citations. For example, “Beyond basic membrane biophysics, there are many simulation studies on bacterial membrane proteins, which require an accurate model for the lipid membrane”. No citations provided.
  • The methods, the abbreviations AMPG, ALPG, and ADPG should be explained (systematic names should be given). Instead of “which is a 14-carbon saturated chain with a methyl group on the second to terminal carbon” also the systematic name should be given. The same is true for surfactants. How was the compressibility modulus calculated?
  • Figure 1 does not shown the chemical structures, but rather the stick models used in the simulations.
  • The simulated systems are not described. The size of the systems is not known. It matters whether it was 128 lipids or 256 lipids. How many surfactant molecules were incorporated into the membranes? Were lipids removed?
  • The results, snapshots of the initial and final configurations of the simulated systems should be shown. The thickness of the membranes can be calculated from Figure 2 and discussed.
  • Table 1, why is the area of BQT (with two tails) almost the same as that for LQT (with one tail)?
  • It is not clear why electric fields were applied to the systems. Such simulations make sense for electroporation (gene transfection). Is electroporation performed for bacteria?
  • How do counterions arrange around the charged lipids and charged surfatants? Can electrostatic charges via counterions alone lead to changes in lipid properties such as APL, thickness and order parameter for the bilayers?
  • This is not very clear to me: the system is simulated with periodic boundary conditions, so the total area of both leaflets must be the same. In the case of asymmetric systems, this leads to an artificial separation of the lipids in the opposite leaflet. In this case, compressibility modulus calculations is unlikely to make sense.

Author Response

This manuscript describes MD simulations of a model bacterial membrane perturbed by small amphiphilic molecules. The authors studied both anionic and cationic surfactants. The work offers new insight into this problem and is of a good technical level. I will recommend its acceptance into Membranes after the following issues have been resolved:

  • The text should be carefully proofread. For example, the Introduction starts from “T The structure and…” which gives a bad impression for further reading.

This was fixed and a careful proof reading was done.

  • The introduction, there are many statements that are not supported by citations. For example, “Beyond basic membrane biophysics, there are many simulation studies on bacterial membrane proteins, which require an accurate model for the lipid membrane”. No citations provided.

We have added additional citations as advised.

  • The methods, the abbreviations AMPG, ALPG, and ADPG should be explained (systematic names should be given). Instead of “which is a 14-carbon saturated chain with a methyl group on the second to terminal carbon” also the systematic name should be given. The same is true for surfactants. How was the compressibility modulus calculated?

We have added the lipid names, added chemical structures for the surfactants and the compressibility modulus.

  • Figure 1 does not shown the chemical structures, but rather the stick models used in the simulations.

We have changed the figure as suggested to include stick models.

  • The simulated systems are not described. The size of the systems is not known. It matters whether it was 128 lipids or 256 lipids. How many surfactant molecules were incorporated into the membranes? Were lipids removed?

We have expanded the methods section to clarify these points, and added a new figure to show the system.

  • The results, snapshots of the initial and final configurations of the simulated systems should be shown. The thickness of the membranes can be calculated from Figure 2 and discussed.

A snapshot of representative systems was added. The bilayer thickness of each system can be clearly seen in Figure 2, and we have added discussion relative to the area per lipid.

  • Table 1, why is the area of BQT (with two tails) almost the same as that for LQT (with one tail)?

Thanks for catching this, we found an error in our FatSlim analysis and have rectified a few of the areas.

  • It is not clear why electric fields were applied to the systems. Such simulations make sense for electroporation (gene transfection). Is electroporation performed for bacteria?

The electric fields were applied to speed-up hydrophilic pore formation. Given that the pores formed through electroporation and at equilibrium are similar in structure, we used the fields to induce pore formation on a simulation time scale. We added this explanation to the results section.

  • How do counterions arrange around the charged lipids and charged surfatants? Can electrostatic charges via counterions alone lead to changes in lipid properties such as APL, thickness and order parameter for the bilayers?

As shown in Figure 5E, the counterion distributions can change substantially, but alone cannot explain even the change in electrostatic potential. We have investigated the ion distributions but did not find any significant trends and leave to future work to fully decompose the interplay between ions and lipids.

  • This is not very clear to me: the system is simulated with periodic boundary conditions, so the total area of both leaflets must be the same. In the case of asymmetric systems, this leads to an artificial separation of the lipids in the opposite leaflet. In this case, compressibility modulus calculations is unlikely to make sense.

This is a good point that we overlooked and therefore we removed the compressibility modulus calculation for the asymmetric systems.

Reviewer 2 Report

The manuscript describes molecular dynamics simulations of symmetric and asymmetric bacterial membranes. I just have a few points for corrections or clarification.

1. Line 28 "T The". Remove the first "T".

2. Fig. 2,. The x axis showing the z axis should be better defined. Is 0 the centre of the membrane? Where is the aqueous solution/membrane interface. Is it at z axis = -3 and +3, or somewhere else? How is z defined? Does a negative value of z indicate the cytoplasmic or the extracellular side of the membrane? In Fig. 2C the value of the electrical potential of > 0.5 V seems to be very high. Are there any experimental values to compare with?

3. Line 97 and lines 405-406. The authors say the including polarisability would be of interest. So does this mean that GROMACS v2018.3 doesn't include polarisability. If so, this should be stated explicitly.

4. Table 2. Three significant figures on a standard error seems too many. For the asymmetric membranes the areas are defined, e. g. 0.856/0.714. What is the value before the slash and what is the value after the slash? I assume that they're individual values for each side of the membrane; but which?

5. I'm not sure what the terms "inner water" and "outer water" mean. I guess that "inner water" might mean the water between the two bilayers and "outer water" might be the water outside the two bilayers. This should be specified more clearly. 

Author Response

The manuscript describes molecular dynamics simulations of symmetric and asymmetric bacterial membranes. I just have a few points for corrections or clarification.

  1. Line 28 "T The". Remove the first "T".

            The T was removed.

  1. Fig. 2,. The x axis showing the z axis should be better defined. Is 0 the centre of the membrane? Where is the aqueous solution/membrane interface. Is it at z axis = -3 and +3, or somewhere else? How is z defined? Does a negative value of z indicate the cytoplasmic or the extracellular side of the membrane? In Fig. 2C the value of the electrical potential of > 0.5 V seems to be very high. Are there any experimental values to compare with?

We added a better definition of the bilayer symmetry and a figure to clarify the system set-up. The electrostatic potential for the CHARMM36 lipids is larger than the experimental estimates, but here we are more focused on the changes to the potential with the addition of the amphiphilic molecules. We are not aware of experiments on the lipids used in our study.

  1. Line 97 and lines 405-406. The authors say the including polarisability would be of interest. So does this mean that GROMACS v2018.3 doesn't include polarisability. If so, this should be stated explicitly.

The CHARMM36 force-field is a fixed charge model. While polarizable lipid force fields exist, they are very expensive, and do not have these types of molecules parameterized. We clarified this point in the discussion.

  1. Table 2. Three significant figures on a standard error seems too many. For the asymmetric membranes the areas are defined, e. g. 0.856/0.714. What is the value before the slash and what is the value after the slash? I assume that they're individual values for each side of the membrane; but which?

We reduced the Sig. Figs. and added and explanation to the table legend regarding the data with a slash.

  1. I'm not sure what the terms "inner water" and "outer water" mean. I guess that "inner water" might mean the water between the two bilayers and "outer water" might be the water outside the two bilayers. This should be specified more clearly. 

We added a figure and explanation to the methods section to clarify the system set-up. This statement was clarified.

Reviewer 3 Report

The manuscript is well written and can be published in the present form. These are routine MD simulations but the results are interesting and the discussion is of interest for the readership of the Membranes journal.

There are some technical issues that have to be addressed before publication: Affiliations, Table 1, Equation on the page 4 etc.

Author Response

The manuscript is well written and can be published in the present form. These are routine MD simulations but the results are interesting and the discussion is of interest for the readership of the Membranes journal.

There are some technical issues that have to be addressed before publication: Affiliations, Table 1, Equation on the page 4 etc.

We have fixed the affiliations, Table 1, and the compressibility equation, and have proof-read again.

Reviewer 4 Report

This paper explores physical properties of the model bacterial membrane (mixed AMPG/LPG/ADPG) with admixtures of (un)charged amphiphilic molecules. Authors, using extensive MD calculations, draw a conclusion that asymmetric load of the amphiphiles perturb membranes more severely compared to symmetric. This is a technically competent contribution, which is, although, not of great interest to the membrane community -- due to trivial conclusions and limited application to the mentioned field (design of antimicrobial molecules).

However, I approve this for publication after a minor revision.

  1. Methods are not complete -- this chapter does not contain any details about electroporation modeling. Which systems were modeled and how many replicas there where; how the pore formation was classified, e.g. -- all this should be provided here.
  2. I suggest that an exhaustive table on simulated systems (composition; parameters; potential; MD duration, etc) would be provided here as well. At the same time, it would summarize the overall MD duration in this work.
  3. Some resulting structures from MD / compactified trajectories should be made accessible -- e.g., uploaded to the Zenodo repository.
  4. Please explain clearly, what is major application of your findings to antibacterial combat?
  5. In table 1, "asym" rows, what are values delimited by the slash?
  6. In Fig. 5B, why do we see a really big shift in the right group of curves (red and yellow vs. green)?
  7. In Table 2, please provide standard deviation.
  8. In Fig. 6A, what are red and blue lines? Also, provide details how the probability (in the raw form) is calculated.
  9. In the Discussion (and maybe Introduction) you probably have to mention and discuss several other works on bacterial and archaeal membranes MD, which analyze heterogeneities and properties change after addition of admixtures (e.g. Lipid II: http://dx.doi.org/10.1038/srep01678) or changing chemical entities of the lipids (e.g. http://dx.doi.org/10.1038/srep07462).

Author Response

This paper explores physical properties of the model bacterial membrane (mixed AMPG/LPG/ADPG) with admixtures of (un)charged amphiphilic molecules. Authors, using extensive MD calculations, draw a conclusion that asymmetric load of the amphiphiles perturb membranes more severely compared to symmetric. This is a technically competent contribution, which is, although, not of great interest to the membrane community -- due to trivial conclusions and limited application to the mentioned field (design of antimicrobial molecules).

However, I approve this for publication after a minor revision.

  1. Methods are not complete -- this chapter does not contain any details about electroporation modeling. Which systems were modeled and how many replicas there where; how the pore formation was classified, e.g. -- all this should be provided here.

The Methods Section was expanded substantially to include these details.

  1. I suggest that an exhaustive table on simulated systems (composition; parameters; potential; MD duration, etc) would be provided here as well. At the same time, it would summarize the overall MD duration in this work.

We do not feel that a table is needed given the expanded methods section in this version.

  1. Some resulting structures from MD / compactified trajectories should be made accessible -- e.g., uploaded to the Zenodo repository.

We have added a sentence that data will be made available upon request.

  1. Please explain clearly, what is major application of your findings to antibacterial combat?

Our major finding is that these molecules greatly effect important membrane properties, particularly the membrane potential difference and the lateral pressure profile, but only when they are asymmetrically distributed. Therefore, when designing novel antibiotics understanding and characterizing how the molecules are distributed and the rate for flip-flop is crucial. We have clarified the discussion to emphasize this point.

  1. In table 1, "asym" rows, what are values delimited by the slash?

We added an explanation in the table legend and a figure showing the set-up.

  1. In Fig. 5B, why do we see a really big shift in the right group of curves (red and yellow vs. green)?

We have changed this figure to adjust the red and yellow curves. The discrepancy is due to different lengths of the water phase in the different systems.

  1. In Table 2, please provide standard deviation.

We have added the standard error to the table.

  1. In Fig. 6A, what are red and blue lines? Also, provide details how the probability (in the raw form) is calculated.

The red/blue lines illustrate the applied electric fields. We have added details of how the probability was calculated to the methods section.

  1. In the Discussion (and maybe Introduction) you probably have to mention and discuss several other works on bacterial and archaeal membranes MD, which analyze heterogeneities and properties change after addition of admixtures (e.g. Lipid II: http://dx.doi.org/10.1038/srep01678) or changing chemical entities of the lipids (e.g. http://dx.doi.org/10.1038/srep07462).

More references were added, including these, and the intro/discussion was expanded.

Round 2

Reviewer 1 Report

The manuscript has been improved.

Reviewer 4 Report

The manuscript may be accepted in the present form.